# Study of SCC of X70 Steel Immersed in Simulated Soil Solution at Different pH by EIS

**DOI:** 10.3390/ma14237445

**Published:** 2021-12-04

**Authors:** Andres Carmona-Hernandez, Ricardo Orozco-Cruz, Edgar Mejía-Sanchez, Araceli Espinoza-Vazquez, Antonio Contreras-Cuevas, Ricardo Galvan-Martinez

**Affiliations:** 1Instituto de Ingeniería, Campus Veracruz, Universidad Veracruzana, S. S. Juan Pablo II, Zona Universitaria, Veracruz 94294, Mexico; andcarmona@uv.mx (A.C.-H.); rorozco@uv.mx (R.O.-C.); araespinoza@uv.mx (A.E.-V.); 2Facultad de Ingeniería, Campus Ixtaczoquitlán, Universidad Veracruzana, KM 1.0 Carretera Sumidero Dos Ríos, Veracruz 94452, Mexico; edmejia@uv.mx; 3Instituto Mexicano del Petróleo, Eje Central Lázaro Cárdenas Norte 152, Ciudad de México 07730, Mexico; acontrer@imp.mx

**Keywords:** X70 steel, pH, stress corrosion cracking (SCC), slow strain rate tests (SSRT), angle phase

## Abstract

An electrochemical study of stress corrosion cracking (SCC) of API X70 steel in a simulated soil solution at different pH values (3, 8 and 10) was carried out. The stress conditions were implemented by slow strain rate stress test (SSRT) and the SCC process was simultaneously monitored by electrochemical impedance spectroscopy (EIS). Fracture surface analysis and corrosion product analysis were performed by scanning electron microscopy (SEM) and X-ray diffraction (XRD), respectively. The results show that the susceptibility to SCC was higher as the pH decreases. In the acid solution, hydrogen evolution can occur by H^+^ and H_2_CO_3_ reduction, and more atomic hydrogen can diffuse into the steel, producing embrittlement. EIS results indicated that the anodic dissolution contributed to SCC process by reducing the charge transfer resistances during the SSRT test. While SEM micrographs shown a general corrosion morphology on the longitudinal surface of samples. At higher pH (pH 8 and pH 10), the SCC susceptibility was lower, which it is attributed to the presence of corrosion products film, which could have limited the process. Using the angle phase (φ) value it was determined that the cracking process started at a point close to the yield strength (YS).

## 1. Introduction

Buried pipelines have been used as one of the most economical and safe ways to transport oil and natural gas to meet energy demand around the world [1]. Nevertheless, it has been known that buried pipelines had experience failures caused by a form of localized corrosion known as stress corrosion cracking (SCC). This is a highly important issue, because always the leakage or rupture and failure of the pipelines can pose a potential threat to humans and environment [2,3,4]. SCC is characterized by initiation and propagation of cracks due to the simultaneous effect of tensile stress and a specific corrosive environment [5]. SCC occurs on the external surface of the steel where the protective coating disbonded and the ground water is in direct contact with the pipeline surface [6]. Based on the pH of the ground water, buried pipelines can experience two main types of SCC, i.e., high pH-SCC (pH > 9) and near neutral pH-SCC (pH 6–8) [7,8,9]. High pH-SCC usually occurs in a concentrated carbonate/bicarbonate electrolyte resulting in an intergranular cracking mode. The mechanism of high pH SCC is attributed to anodic dissolution resulting from selective dissolution at the grain boundary and repeated rupture of passivating film that form over the crack tip [10]. On the other hand, the occurrence of near-neutral pH SCC is associated with a special environment characterized with anaerobic, dilute bicarbonate solutions and the stress corrosion crack has a transgranular, quasi-cleavage crack morphology [11,12,13]. Some researchers [14,15] believed that near-neutral pH SCC is dominated by the mechanism of hydrogen-facilitated dissolution. In addition, other authors [16,17,18] have investigated the SCC behavior of pipelines in acidic soil environments (pH 3–6), in which both hydrogen embrittlement and anodic dissolution contribute to the SCC process. Therefore, it is evident that solution pH is one of the environmental parameters that has the greatest effect on SCC of buried pipelines. Chen et al. [19] studied the SCC behavior of X70 steel in several soils and reported that the steel was more susceptible as the soil solution pH increased in the range of 5.5–7. Liu et al. [20] investigated the SCC process of a X70 steel varying the pH (from 3.5 to 6.5) of a synthetic acidic soil solution and at different cathodic potentials. They found that depending on pH and cathodic potential value, the SCC mechanism can be among anodic dissolution, hydrogen embrittlement or the combination of both. Gadala and Alfantazi [21] investigated the effect of pH in a range of 4 to 9 on the corrosion of X100 steel in a simulated soil solution. Their electrochemical measurements showed that corrosion rates of the steel decreased with pH and revealed the involvement of adsorption and parallel adsorption–diffusion in near-neutral and mildly alkaline conditions, respectively. Cui et al. [22] studied the SCC behavior of X70 pipeline steel in dilute bicarbonate solutions at several pH (4–6.8). Their result indicated that when the solution pH decreased from 6.8 to 5.5, SCC susceptibility decreased because of the increment of the anodic dissolution; and when the solution pH decreased from 5.5 to 4.0, SCC susceptibility increases due to the enhancement of cathodic reactions. Liang et al. [23] investigated the corrosion and SCC behavior of X80 steel in alkaline soil solution with different pH (8, 10 and 11) and found that the general corrosion and SCC resistance increased with increasing pH, which they attributed to formation of a protective FeCO_3_ film in the steel surface. Despite mentioned above, the effect of soil solution pH on the SCC behavior of pipeline steels is not well defined and more research should be done on this issue. An important point to note is that most of the works concerning the effect of pH on the SCC behavior in simulated soil solution was performed electrochemical measurements in unstressed samples [20,21,22,23]. Therefore, it is important to carry out these measurements when the sample is under stress to obtain electrochemical data during the SCC process. In this sense, electrochemical impedance spectroscopy (EIS) is an electrochemical technique that has been used to study passivation breakdown, crack initiation and propagation, and other events that take place during the SCC process on different metals [24,25,26,27]. Some studies [28,29,30] have carried out EIS measurements simultaneously on unstressed and stressed specimens, and they have suggested that the evolution of the phase angle shift (φ) at specific frequencies may be related to the initiation and propagation of cracks. On the other hand, researchers [28,31,32,33] have related changes in the low frequency region of the EIS Nyquist plots with the formation of cracks and have developed complex equivalent circuit models to simulate crack geometry changes and crack tip/wall electrochemical activity changes during cracking process. 

The aim of the present work was to assess the pH effect on electrochemical and SCC behavior of X70 steel immersed in a simulated soil solution using EIS measurements, slow strain rate test (SSRT) and surface analysis techniques (SEM).

## 2. Materials and Methods

### 2.1. Material and Test Solution

X70 steel samples were obtained from a pipeline steel with external diameter of 36 in (914.4 mm) and a wall thickness of 0.902 in (22.91 mm). Smooth cylindrical tensile samples of API 5 L X70 were made from pipeline in cross direction, and they were machined according to the NACE TM-198 standard [34]. Transverse direction was chosen as the transversal stresses are more critical than longitudinal stresses in a pipeline. Table 1 shown the chemical composition of X70 steel.

A synthetic soil solution known as NS4 as test solution was used. NS4 solution has the following composition: 0.483 g/L NaHCO_3_, 0.122 g/L KCl, 0.137 g/L CaCl_2_, and 0.131 g/L MgSO_4_ 7H_2_O. Several studies [35,36,37] have used the NS4 solution to simulate the ionic species present in the soil electrolyte.

NS4 solution was prepared from analytical grade reagents with distilled water at different pH: 3, 8 and 10. The solution pH was adjusted by using dilute solutions of NaOH or HCl. The solution test was employed under aerated conditions at atmospheric pressure and room temperature. 

### 2.2. Slow Strain Rate Test (SSRT)

The SSRT test were conducted in a constant extension rate tests machine (M-CERT, Intercorr International, Houston, TX, USA) with load capacity of 44 kN and total extension of 50 mm at strain rate of 1 × 10^−6^ s^−1^. A schematic representation of the SSRT setup is illustrated in Figure 1. The SSRT tests were carried out both in air and NS4 solution at different pH values at room temperature. Before each test, tensile samples were polished (SiC sandpaper up to 1200 grit) in the gauge section along the direction vertical to avoid small defects and superficial damages. All the specimens were rinsed with distilled water and degreased with acetone, and finally dried in air. The exposed area of specimen was 2.84 cm^2^. 

After SSRT tests, the fracture surface of specimens was cleaned with inhibit acid containing 1000 mL HCl, 20 g Sb_2_O_3_ and 50 g SnCl_2_ as per the guidelines provided in ASTM G1-03 [38]. Then the fracture surface and longitudinal surface of the specimen were observed by scanning electron microscopy at an accelerating voltage of 5 kV and vacuum pressure of 9.6 × 10^−5^ Pa. X-ray diffraction analysis was also conducted to analyze the composition corrosion products formed on the steel surface.

### 2.3. Electrochemical Impedance Spectroscopy (EIS)

EIS measurements were carried out on unstressed samples and stressed samples during SSRT test, using a potentiostat/galvanostat. As illustrated in Figure 1, a typical three-electrode configuration electrochemical cell was used, where the X70 steel acted as the working electrode, saturated calomel electrode (SCE) as reference electrode and a graphite rod as auxiliary electrode. EIS data were collected under free potential over a frequency range from 10,000 to 0.01 Hz, seven points per decade of frequency were recorded with an amplitude of 0.01 V vs. E_corr_. The EIS data were fitted by an open available software and Kramers–Kronig Transform (KKT) was used to check the validity of the experimental EIS data [39]. All experiments were performed in triplicate.

## 3. Results and Discussion

### 3.1. Slow Strain Rate Test (SSRT)

Stress-strain curves obtained from SSRT for X70 steel in air and NS4 solution at different pH along with different points indicated on a stress-strain curve for monitoring EIS are shown in Figure 2. T0 represents a point at beginning of the SSRT test, EZ at elastic zone, YS at yield strength, UTS at ultimate tensile strength and BF before fracture. In addition, mechanical parameters obtained from SSRT curves are shown in Table 2. The stress-strain curves in the NS4 solution reflected the degradation of some mechanical properties of the steel compared to those obtained in air. The percentage of elongation (%η) was lower as the pH of the solution decreased. Cui et al. [22] reported a similar behavior of %η in the same solution, however the pH range (4–6.8) was different from the present study. Also, the elastic modulus (E_σ_) of the steel decreased in the test solution in comparison to that obtained in air. Some SCC research of pipeline steel in NS4 solution [40,41] have presented a decreased E_σ_ value. Likewise, at low pH, E_σ_ decreased the stiffness of the specimen; this can be attributed by the combination of corrosion activity on the metal surface and the effects that generate the H diffusion into the metal crystal lattice provoking the weakening of the interatomic bonds. On the other hand, the yield strength (σ_YS_) and ultimate tensile strength (σ_UTS_) values decreased at pH 8 and pH 10 in comparison with air condition, while at pH 3, this tensile strength increased due to the embrittlement of the material. This increase in σ_YS_ and σ_UTS_ and loss in ductility has been reported by other authors [37,42,43], which can be attributed to the hydrogen diffusion into the crystal lattice of metal and interacts with the dislocations and blocks their movement, which produce hardening of the steel. In the acid solution (pH 3), atomic hydrogen is generated by the cathodic reaction of H^+^ protons [43]. Liu et al. [44] point out that when the concentration of hydrogen in steel is not sufficient and below a critical value, hydrogen could impede the slip of dislocations, leading to an increase of steel yield strength. 

The degree of susceptibility to SCC of X70 steel in NS4 solution at different pH values was assessed based on NACE TM-0198 [34]. Susceptibility to SCC was expressed in terms of the reduction in area index (I_Ψ_) and plastic elongation index (I_ε_), according to following equations: (1)$$\Psi_{\left(S,0\right)}=\left(D_{i}^{2}-D_{f}^{2}\right)/D_{i}^{2}\;$$
(2)$$I_{\Psi }=\Psi_{S}/\Psi_{0}\;$$
where Ψ is the percentage reduction in area, D_i_ and D_f_ are the initial and final diameter of the fracture surface, respectively.
(3)$$\varepsilon_{p\left(S,0\right)}=\left[\frac{\eta_{f}}{L_{i}}-\left(\frac{\sigma_{f}}{\sigma_{\mathrm{PL}}}\right)\left(\frac{\eta_{\mathrm{PL}}}{L_{i}}\right)\right]$$
(4)$$I_{\varepsilon }=\frac{\varepsilon_{\mathrm{pS}}}{\varepsilon_{p0}}$$
where ε_p_ is the plastic elongation, L_i_ is the initial gauge length, σ_PL_ and η_PL_ are the stress and elongation at proportional limit, respectively, whereas σ_f_ and η_f_ are the stress and elongation at failure, respectively. The suffix 0 and S correspond to the values obtained in air and the NS4 solution, respectively.

Based on the I_Ψ_ and I_ε_ values summarized in Table 2, and to the classification proposed by McIntyre et al. [45], the X70 steel exhibited a moderate susceptibility to SCC in the NS4 at pH 3, whereas the steel had low susceptibility to SCC at pH 8 and pH 10. 

### 3.2. SEM Observations

Figure 3 shows the SEM micrographs of fracture surfaces of the X70 steel after SSRT tests in air and the NS4 solution at different pH values. For the test in air (Figure 3a), the fracture surface showed a typical ductile fracture mode (cup-and-cone) with large plastic deformation. In Figure 3a1, many cavities produced by micro-plastic deformation and microvoids were observed, which could act as stress concentrators being these sites the preferred sites for crack nucleation. According to Contreras et al. [35] these microvoids produce metal failure by coalescence mechanism. As for the X70 steel in NS4 solution, an increase in the fracture area was observed as decreased the pH value (Figure 3b–d). In the acid solution (Figure 3b), the steel exhibited transgranular and brittle type of fracture at macroscopic level. Nevertheless, the microscopic fracture morphology displayed dimpled appearance typical of ductile fracture as shown in Figure 3b1. Lynch [46] suggested that this type of fracture morphology can be produced by a hydrogen embrittlement mechanism known as adsorption-induced dislocation emission (AIDE), which consists of the weakening of the interatomic bonds in the crack tip by adsorbed hydrogen, thus promoting fracture at a relatively lower strain due to local reduction in ductility. One of fracture morphology features produced by AIDE is that dimples appear to be smaller and shallower than those produced by ductile fracture in air. However, comparison of dimple size is difficult because of the large, deep dimples produced by fracture in the absence of hydrogen; stretched dimples are often difficult to differentiate. Wang et al. [47] also obtained a similar fracture morphology in hydrogen pre-charged X70 steel samples. It is important to point out that in Figure 3b, internal cracks were observed in the fracture surface. These cracks can be related to hydrogen diffusion into the material and its trapping at dislocations, which causes weakening of interatomic bonds and facilitates decohesion [48,49]. On the other hand, the steel fracture in the NS4 solution at pH 8 (Figure 3c,c1) was ductile with a dimpled morphology. Finally, in the alkaline solution (Figure 3d,d1), the fracture surface morphology was similar to that obtained in air, with a predominant necking phenomenon and ductile dimple fracture mode. 

It has been acknowledged that the presence of secondary cracks on the material surface is a key sign of SCC occurred [32,50]. SEM micrographs of the longitudinal surfaces of X70 steel samples in the necked region of the gage section are shown in Figure 4. In the NS4 solution at pH 3 (Figure 4a), some secondary cracks with apparent branching features were observed. From Figure 4a1, it is seen that there is a rough surface around the cracks, which is indicative a severe corrosion attack. One important point to note is that the secondary cracks can initiate by the corrosion pits on the steel surface by anodic dissolutions, which can turn into cracks based on the localized stress concentration effect, pit geometry and local environment that can develop inside the pits. In addition, this corrosion morphology can initiate in metallurgical defects as inclusions. Wang et al. [51] claimed that Si-enriched inclusions can promote the initiation of microcracks on the X70 steel in the NS4 solution. So, the hydrogen embrittlement in synergy with the pitting corrosion generates secondary cracks and according to the characteristic of these crack (crack depth and crack length) the steel ductility can decrease. Liu et al. [52] point out that the ductility parameter Ψ decreased as the cracks and defects increased in the steel; this fact is due to microcracks coalescence to form the main crack leading to final fracture, exhibiting the material a loss of ductility. In contrast, in the NS4 solution at pH 8 (Figure 4b) [53] and pH 10 (Figure 4c), the secondary cracking was minimal, and the magnified SEM micrographs of the secondary cracks (Figure 4b1,c1) revealed no sign of corrosion attack and shallow marks of the extensive plastic deformation. It is important to point out that despite that the steel exhibit a low susceptibility at these pHs, a few secondary cracks were observed close to the neck region. Similar results were reported by Fan et al. [54]. In their study, an X80 steel showed a low susceptibility to SCC in a carbonate-bicarbonate solution without chloride ion, but shallow and short secondary cracks initiated on the lateral surface. Finally, the number of the secondary cracks increased, and cracks became longer and deeper as the solution pH value was lower (Figure 4a1–c1). The above confirmed that the SCC susceptibility of the X70 steel increased with decreasing pH. 

### 3.3. Electrochemical Impedance Spectroscopy (EIS)

Figure 5 shows the Nyquist and Bode plots of X70 pipeline steel immersed in NS4 solution at under different pH values. EIS plots were recorded on different points of the stress-strain curve, which are indicated in Figure 5. At pH 3 (Figure 5a), Nyquist curves shown a capacitive loop at the high-frequency, which reflects an electrode reaction at the interface and an electric double-layer process, and an inductive loop in the low-frequency range, which was related to the hydrogen evolution and to the relaxation of intermediates adsorption species (FeOH_ads_) formed during the active dissolution of steel [55]. At the pH 8 and pH 10 (Figure 5c,e), when the steel was subjected to elastic stress (points T0 to YS) the Nyquist curves showed only a capacitive loop, i.e., a single time constant attributed to charge transfer process. Subsequently, at the UTS and BF points, a second capacitive loop at high frequencies appeared, this fact is associated with the reactions that take place on a corrosion products’ film [26,56]. 

From Bode plots, at pH 3 (Figure 5b), a single peak was observed, indicating a single time constant (τ). At pH 8 and pH 10 (Figure 5d,f), bode plots showed two peaks at the UTS and BF points, due to the presence of corrosion products film. These results were similar to those reported by Carpintero et al. [26]. On the other hand, at pH3 (Figure 5b), the maximum phase angle (φ_max_) decreased during the SSRT test except at BF point, whereas at pH 8 and pH 10 (Figure 5d,f), φ_max_ significantly decreased at UTS and BF points. Lou & Singh [31] suggested that an increase in corrosion activity at the crack tip serve as a main reason for the significant drop in φ_max_ during SSRT. Finally, in Bode plots, it is observed that KK transformed had well data matched with the experimental values over almost the entire frequency range, except at the high frequencies (10^4^–10^3^ Hz). Some authors [57,58] can suggest a measurement frequency range of 1000–0.1 Hz that it ensure stable system conditions. 

For an approximate physical description of the EIS measurements of the steel in NS4 solution at different pH during SSRT test, equivalent electric circuits (EEC) are suggested as shown in Figure 6. The first EEC (Figure 6a) was employed to fit the EIS data in the acid solution (pH 3), in which the R_s_ is solution resistance, R_ct_ is the charge-transfer resistance, L_ads_ and R_ads_ are the inductance and resistance of the adsorption process, respectively [22,54]. The constant phase element (CPE_dl_) is an empirical circuit element that represents the non-ideal capacitive behavior of the electrode. The impedance of CPE (Z_CPE_) is calculated by following equation [59]:(5)$$Z_{\mathrm{CPE}}=Y_{0}{\left(\omega j\right)}^{-n}$$
where j is the imaginary number, Y_0_ stand for the CPE constant, ω is the angular frequency and n is the CPE exponent which is regarding to the surface inhomogeneity degree [60]. The double layer capacitance (C_dl_) values were determined from the Equation (6) given by Brug et al. [61].
(6)$$C_{\mathrm{dc}}=Y_{0}{}^{\frac{1}{n}}{\left(R_{s}{}^{-1}+R_{\mathrm{tc}}{}^{-1}\right)}^{\frac{n-1}{n}}$$

On the other hand, the EEC of Figure 6b was used to fit the EIS data from T0 point to YS point, and the EEC of Figure 6c to the UTS and BF points both at pH 8 and pH 10. The parameters R_f_ and CPE_f_ are the resistance and non-ideal capacitance of the corrosion products film. 

The electrochemical parameters obtained from fitting of EIS spectra are summarized in Table 3, and the behavior of R_ct_ and C_dl_ values during SSRT are illustrated in Figure 7. Figure 7 and Table 3 show that at pH 3, the R_ct_ decreased from the T0 point to the UTS point. This R_ct_ decrease can be explained by the fact that the gradual stress increment enhances the anodic dissolution rate of the steel. In addition, the diffusion of H atoms, obtained by the reduction reaction, increases because the plastic deformation and the concentration of dislocations at the crack tip, could also enhance the anodic dissolution rate. Cheng [14] proposed in his research that occurrence of SCC depends on the synergism of stress, hydrogen, and anodic dissolution at the crack tip of the steel. On the other hand, the increase of R_ct_ at BF point can be because the crack blunting takes place due to final microvoid formation and coalescence [31]. In the case of pH 8 and pH 10, the behavior and magnitude of R_ct_ during the SSRT test were similar at both pHs. The R_ct_ increased from the T0 point to the EZ or YS point, and then decreased at later points. This behavior of R_ct_ is attributed to the fact that when the steel is subjected to elastic stress, some dislocations oscillate around their equilibrium positions, and the electrochemical behavior at the interface is not significantly altered, thus allowing the formation of the corrosion products film. However, once the local stress exceeds the yield strength, significant movement of the dislocations occurs, resulting in the formation of emergent sites and slip steps on the steel surface, which they can introduce local active sites that can accelerate the localized corrosion process by breaking of corrosion products film and initiation of cracks [53,62].

At all pHs, the C_dl_ value increased throughout the SSRT test. This increase could be related to generation of a new active surface, i.e., increase in exposed area caused by the opening of the main crack and the appearance of secondary cracks. On the other hand, the porous formation and conductive corrosion products film on the surface can also produce an increase in C_dl_ values [63]. It is well known that an increase in solution pH favors the formation of corrosion products on the steel surface [64,65]. Therefore, at pH 8 and 10, the corrosion products could have produced an increase in the C_dl_ value. 

#### Determination of SCC Crack Initiation from the Phase Shift (Δφ) Value

The evolution of phase shift Δφ value at different frequencies test for X70 steel immersed in NS4 solution with different pH values during the SSRT is presented in Figure 8. Δφ is the subtraction between the phase angle of the unstressed specimen and the phase angle of the stressed specimen [28]. It should be point out that Δφ values showed high instability at frequencies of 0.1 and 0.01 Hz. For this reason, only frequencies at the range of 100, 10 and 1 Hz were shown in Figure 8. In acidic solution (Figure 8a), a sharp increase of Δφ value occurred after 6 h of exposure (point close to the yield strength) at a frequency of 10 Hz, whereas the NS4 solutions at pH 8 and pH 10 (Figure 8b,c), this increase of Δφ value happened after 6–8 h of exposure at a frequency of 1 Hz. According to literature [66], the stress level necessary to produce SCC is below the macroscopic σ_YS_ value of the material (stresses range between 85 to 95% of σ_YS_), enough stresses to induce localized microplastic deformation. This microplastic deformation is a key element in the crack initiation and crack propagation process. Therefore, the increase of Δφ value at frequencies between 1 to 10 Hz can be related to initiation and propagation of SCC on X70 steel in the NS4 solution, even at pH 8 and pH 10, conditions in which steel had low susceptibility to SCC. Other research [32,53] also detects SCC crack initiation at this frequency range. 

### 3.4. X-ray Diffraction (XRD) Analysis

Figure 9 shows the XRD diffractograms of corrosion products collected from the X70 steel after exposure in NS4 solution at different pH values. At pH 3, the major phase identified was magnetite (Fe_3_O_4_), and small content of lepidocrocite (γ-FeOOH) and goethite (α-FeOOH). Liu et al. [67] reported that the corrosion products formed on X80 steel in acidic soils (pH 4.5) were FeOOH, Fe_2_O_3_, and Fe_3_O_4_. Tamura [63] point out that when the rate of anodic dissolution is high, a thick film of Fe(OH)_2_ can be formed, which when oxidized directly with air can form Fe_3_O_4_. On the other hand, at pH 8, the main phases were γ-FeOOH and α-FeOOH, which are commonly formed on steel in neutral or alkaline solutions [61]. Jegdić et al. [68] mentioned that γ-FeOOH is semi-conductive and electrochemically active, whereas α-FeOOH is thermodynamically stable and has protective characteristics. These corrosion products can be formed by dehydration of the Fe(OH)_3_. Finally, in the NS4 solution at pH 10, the phases found in the corrosion products were α-FeOOH and maghemite (γ-Fe_2_O_3_). According to literature [69,70] γ-Fe_2_O_3_ is an oxide that regularly forms part of the passive film of steel. Therefore, the formation of α-FeOOH and γ-Fe_2_O_3_ at the interface could appear as the time constant at high frequencies in the EIS plots at pH 8 and pH 10, and this fact was evidenced by the significant increase in C_dl_ and the R_s_ because they were corrosion products with semi-protective characteristics. 

### 3.5. Effect of pH Solution on Electrochemical Reactions

As observed in EIS plots (Figure 5), the electrochemical behavior of X70 steel during SSRT is different depending of the pH value. Regardless of the pH value, the anodic reaction of X70 steel in NS4 solution is mainly the dissolution of iron (7) [11]:(7)$$\mathrm{Fe}\to {\mathrm{Fe}}^{2+}+2e^{-}$$

The cathodic reactions that can occur on the steel surface depend on the pH. At pH 3, the main cathodic reactions are as follows:(8)$$O_{2}+4H^{+}+4e^{-}\to \;2H_{2}{O\;\;\;\;E}_{O_{2}}^{\mathrm{eq}}=760\;\mathrm{mV}\;\mathrm{vs}.\;\mathrm{SCE}\;$$
(9)$$2H^{+}+2e^{-}\;\to {\;H}_{2}{\;\;\;\;\;E}_{H^{+}}^{\mathrm{eq}}=-417\;\mathrm{mV}\;\mathrm{vs}.\;\mathrm{SCE}\;$$

The equilibrium potentials (E_eq_) were calculated based on solution chemical composition and the acid dissociation constants (pKa) using the Nernst equation [71,72]. In addition, the NS4 solution contains HCO_3_^−^ and H_2_CO_3_ species that, under certain conditions, can contribute to the SCC process. The presence of H_2_CO_3_ is due to the hydrolysis of HCO_3_^−^, and a lower pH corresponds to a higher H_2_CO_3_ concentration in the solution. Therefore, the direct reduction of H_2_CO_3_ can occurs according to the following reaction [16,22]:(10)$$2H_{2}{\mathrm{CO}}_{3}+2e^{-}↔H_{2}+2{\mathrm{HCO}}_{3}^{-}{\;\;\;E}_{H_{2}{\mathrm{CO}}_{3}}^{\mathrm{eq}}=-416\;\mathrm{mV}\;\mathrm{vs}.\;\mathrm{SCE}\;$$

According to the E_eq_ values, the reactions (8), (9) and (10) can occur simultaneously on the surface of the steel, since the E_corr_ value in the NS4 solution at pH 3 was approximately −650 mV vs. SCE [19], being more negative than E_eq_ values. Therefore, the atomic hydrogen produced by the reactions (9) and (10) can be absorbed on the surface, as seen at the low frequency range in the Nyquist curves (Figure 5a) and diffuse into the steel preferentially concentrating in sites with high triaxial stresses, such as the tip of a crack and in turn enhanced the anodic dissolution at the crack tip, contributing to the propagation of cracks by SCC [41]. 

In the case of pH 8 and pH 10, the main cathodic reaction was the reduction of O_2_ according to the following reaction [11]:
(11)$$O_{2}+2H_{2}O+4e^{-}\to \;2H_{2}O$$

However, at an alkaline pH, the concentration of HCO_3_^−^ and CO^3-^ can be significant. Liu, et al. [16] claimed that the reduction of HCO^3−^ (12) could be carried out according to the following reaction:(12)$$2{\mathrm{HCO}}_{3}^{-}+2e^{-}↔H_{2}+2{\mathrm{CO}}_{3}^{-}{\;\;E}_{{\mathrm{HCO}}_{3}^{-}}^{\mathrm{eq}}=-713\;\mathrm{mV}\;\left(\mathrm{pH}\;8\right)\;\&-831\;\mathrm{mV}\;\mathrm{vs}.\;\mathrm{SCE}\;\left(\mathrm{pH}10\right)\;$$

Nevertheless, based on the E_eq_ values, the reaction probably would not take place at both pH, since the E_corr_ value varied approximately between −630 and −735 mV vs. SCE [53], being less negative than E_eq_ values. It is important to point out that as the reduction of O_2_ is the main cathode reaction at these pHs, a local acidic environment could be developed by an effect of differential aeration mechanisms inside corrosion pits [73], provoking the initiation of the fewer cracks that were observed by SEM micrographs. However, the role of hydrogen in the SCC process was negligible in comparison to acidic solution. 

## 4. Conclusions

Considering the results of this study about the effect of solution pH on the SCC susceptibility of X70 immersed in a simulated soil solution, several conclusions can be drawn: According to the SSRT results, the SCC susceptibility increase as the pH value decreases. At pH 3, the X70 steel was moderately susceptible to SCC, whereas at pH 8 and pH 10, the X70 steel presented low susceptibility to SCC. Fractographic analysis by SEM revealed that at pH 3, the fracture surface exhibited a brittle appearance with presence of internal cracks, indicating that hydrogen was involved in the SCC process. On the other hand, at pH 8 and pH 10, the steel exhibited extensive plastic deformation and ductile fracture mode. Longitudinal surface of samples showed that the secondary cracking was more severe as solution pH decreased. EIS measurements allowed evaluating the effect of stress and changes in the metal-solution interface by monitoring parameters such as R_ct_ and C_dl_ during the SSRT tests. EIS results indicated that both anodic dissolution and hydrogen embrittlement contributed to the SCC of the steel in the acid solution. Finally, the evolution of Δφ at frequencies below 10 Hz during SSRT test was able to detect the crack initiation by SCC at a closed point to the yield strength of the steel.XRD results showed that the solution pH changed the components of the corrosion products film, which limited the SCC process in the NS4 solution at pH 8 and pH 10.

## Figures and Tables

**Figure 1 materials-14-07445-f001:**
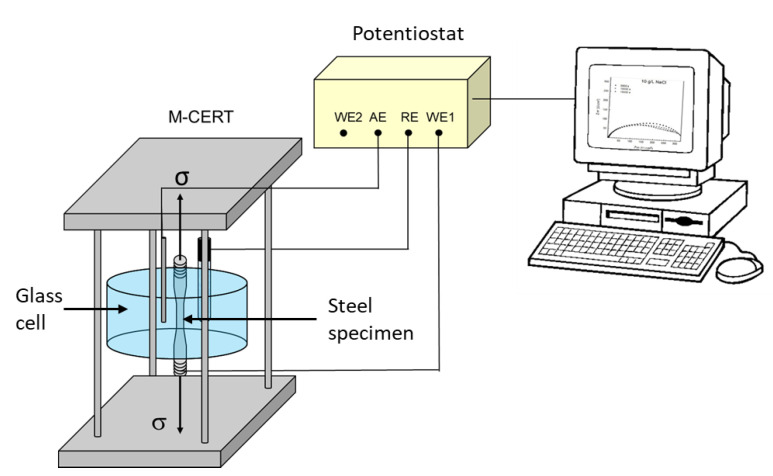
Schematic representation of SSRT setup and electrochemical cell for impedance measurements.

**Figure 2 materials-14-07445-f002:**
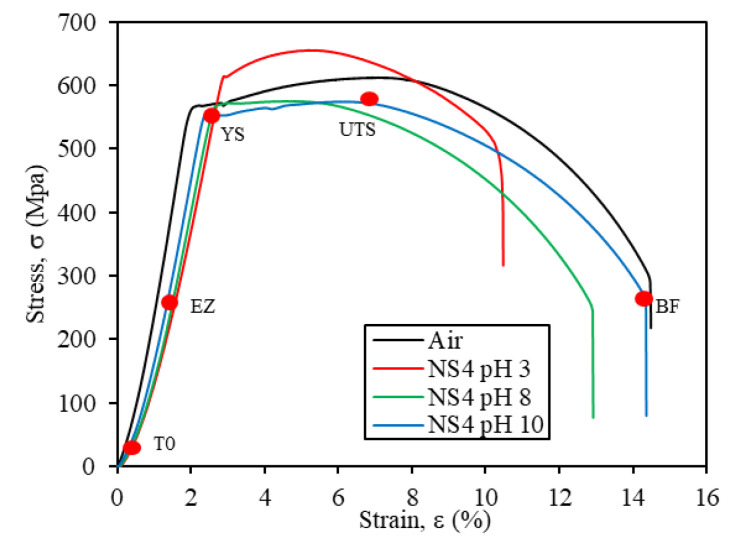
Stress-strain curves of X70 steel obtained in air and in the NS4 solution at different pH values.

**Figure 3 materials-14-07445-f003:**
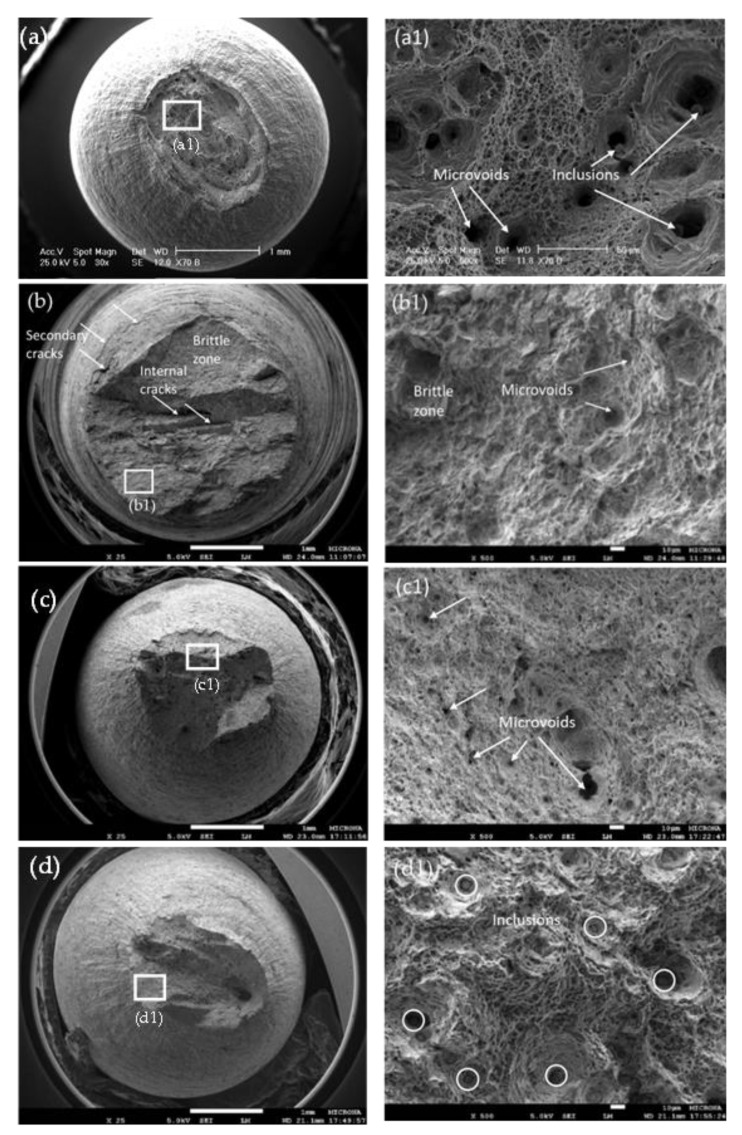
SEM micrographs of fracture surfaces of the X70 steel after SSRT tests in air (**a**,**a1**) and the NS4 solution at different pH values: (**b**,**b1**) pH 3 (**c**,**c1**) pH 8 and (**d**,**d1**) pH 10.

**Figure 4 materials-14-07445-f004:**
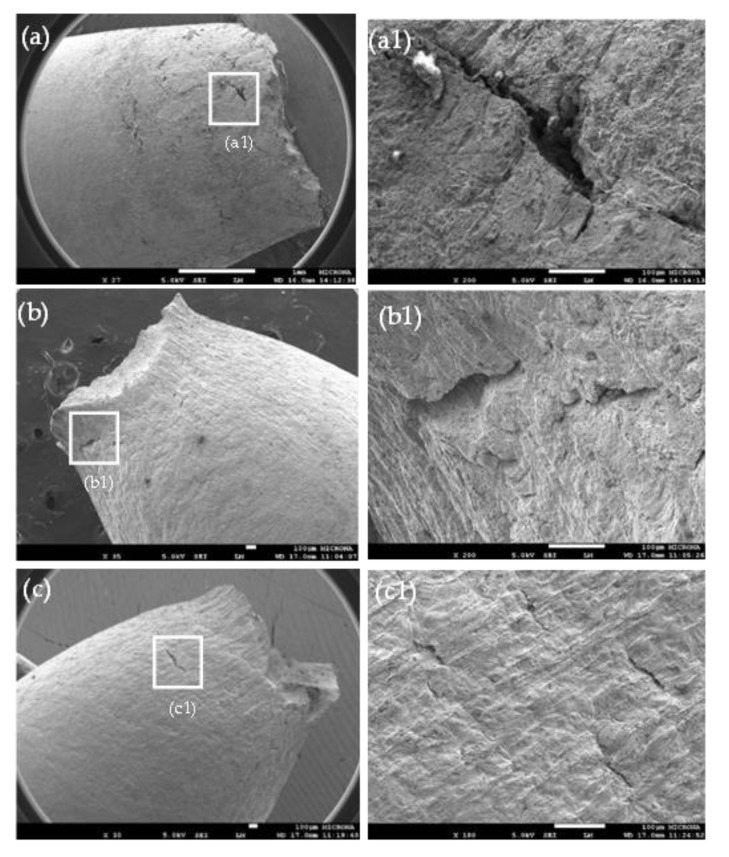
SEM micrographs of side surfaces of X70 steel in the NS4 solution at different pH, (**a**,**a1**) pH 3 (**b**,**b1**) pH 8 and (**c**,**c1**) pH 10.

**Figure 5 materials-14-07445-f005:**
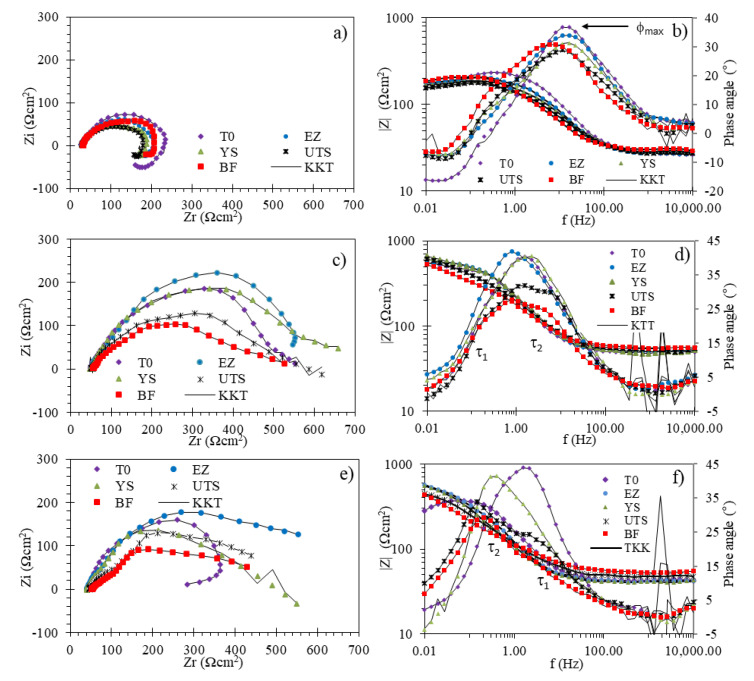
Nyquist (**a**,**c**,**e**) and Bode (**b**,**d**,**f**) plots for X70 steel in NS4 solution at different pH during SSRT test: pH 3 (**a**,**b**), pH 8 (**c**,**d**) and pH10 (**e**,**f**).

**Figure 6 materials-14-07445-f006:**
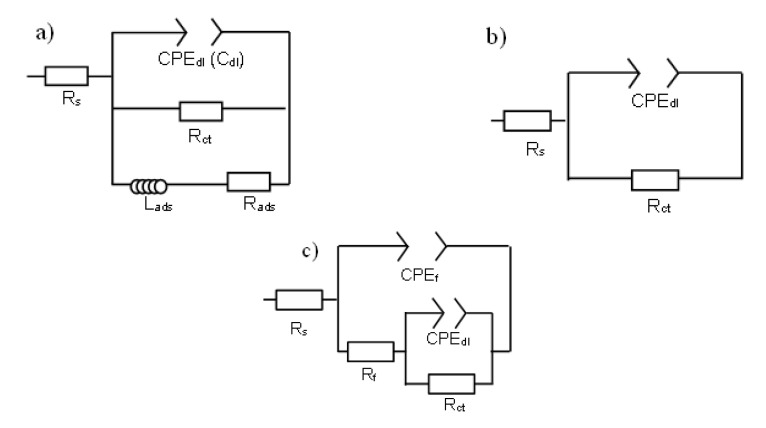
EEC for X70 steel in the NS4 solution at different pH: (**a**) pH 3, (**b**) pH 8 and (**c**) pH 10.

**Figure 7 materials-14-07445-f007:**
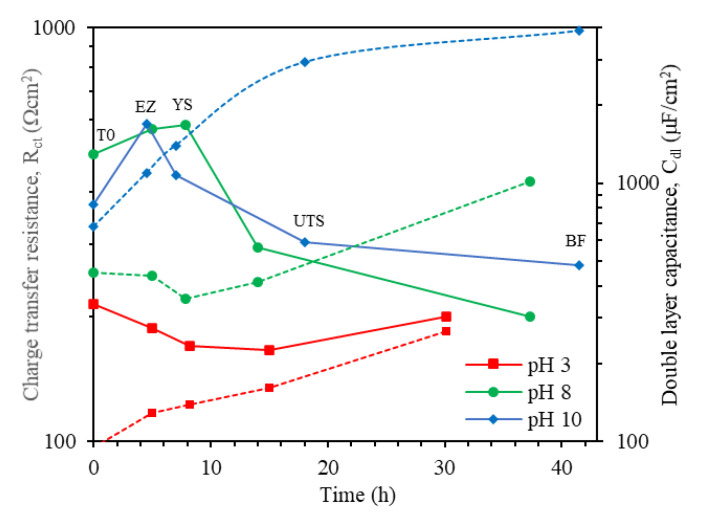
Charge transfer resistance, R_ct_ (solid lines) and double layer capacitance, C_dl_ (dashed lines) values for X70 steel exposed to NS4 solution at different pH values during SSRT test.

**Figure 8 materials-14-07445-f008:**
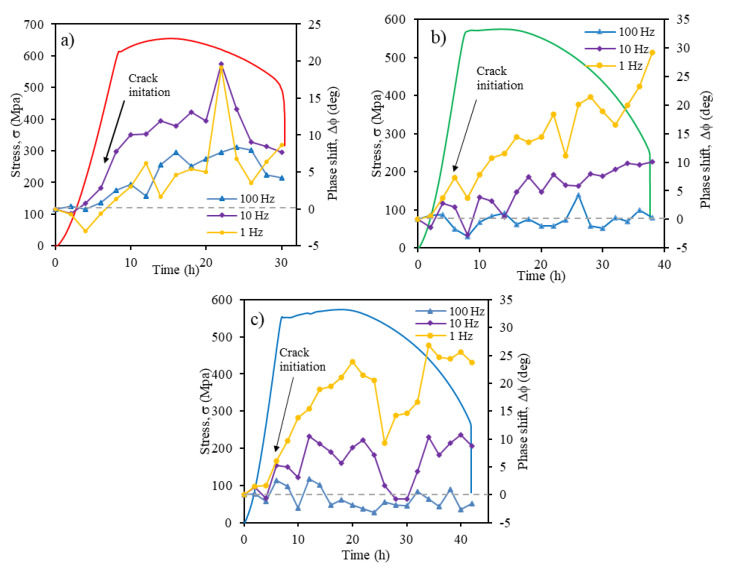
Evolution of Δφ value obtained during SSRT at different frequencies (1, 10 and 100 Hz) for X70 steel in NS4 solution at different pH values: (**a**) pH 3, (**b**) pH 8 and (**c**) pH 10.

**Figure 9 materials-14-07445-f009:**
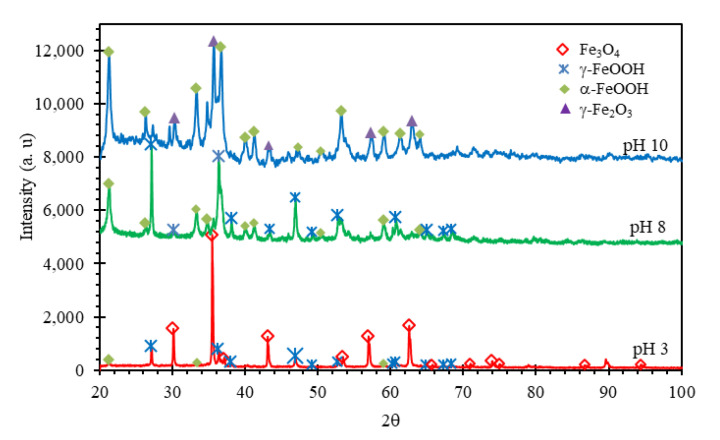
XRD patterns of corrosion products obtained from X70 steel after exposure in NS4 solution at different pH values.

**Table 1 materials-14-07445-t001:** Chemical composition of API X70 steel (wt.%).

Element	C	Mn	Si	Al	Nb	Cu	Cr	Ni	V	Ti	Fe
wt.%	0.03	1.48	0.13	0.033	0.1	0.29	0.27	0.16	0.004	0.012	Bal.

**Table 2 materials-14-07445-t002:** Mechanical properties obtained from the SSRT test.

Condition	TF ^1^ (h)	$E_{\sigma }$ (GPa)	$\sigma_{YS}$ (MPa)	$\sigma_{UTS}\;(\mathrm{MPa})$	η (%)	Ψ (%)	$\varepsilon_{p}$ (%)	$I_{\Psi }$	$I_{\varepsilon }$
Air	42.2	344	566	612	14.5	84	0.13		
NS4 pH 3	30.4	279.4	604.7	655.3	10.4	62.8	0.09	0.74	0.64
NS4 pH 8	37.6	285.2	564	575.2	12.9	78.3	0.12	0.94	0.88
NS4 pH 10	41.9	299.4	552	574.4	14.4	81.8	0.13	0.96	0.99

^1^ TF = Time to failure.

**Table 3 materials-14-07445-t003:** EIS parameters obtained by fitting data to the EEC from Figure 6.

pH	Point	R_s_ (Ω cm^2^)	R_f_ (Ω cm^2^)	CPE_f_-Y_0_ (s^n^/Ω cm^2^)	n_f_	C_dl_ (μF/cm^2^)	R_ct_ (Ω cm^2^)	R_ads_ (Ω cm^2^)	L (H/cm^2^)
3	T0	26.5				94.5	214.4	232.96	1882.4
	EZ	26				128.4	188.2	487.04	3349.5
	YS	27.2				138.3	170.1	465.93	4253
	UTS	26.4				161.2	166.2	337	3037.3
	BF	29.2				265.1	200.1	483.55	4786.3
8	T0	48.7				449.5	494.32		
	EZ	50.1				436.9	567.9		
	YS	48.6				355.4	583		
	UTS	51.8	175.92	9.9 × 10^−4^	0.74	414.4	293.8		
	BF	55.2	178.69	1.5 × 10^−3^	0.74	1014	200		
10	T0	42.5				677.2	373.58		
	EZ	40.8				1090.7	584		
	YS	41.2				1399	440.22		
	UTS	47.4	127.44	2.0 × 10^−4^	0.72	2945.3	303.11		
	BF	53.4	91.106	2.4 × 10^−3^	0.67	3896	266.62		

## Data Availability

The data presented in this study are available on request from the corresponding author.

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
