# Peer review of "Study of SCC of X70 Steel Immersed in Simulated Soil Solution at Different pH by EIS"

_materials, 2021, doi:10.3390/ma14237445_

Round 1

Reviewer 1 Report

In this manuscript, the stress corrosion cracking behavior of X70 pipeline steel was studied as a function of pH in a soil simulated solution. The manuscript is well written and interesting findings were presented. There are some comments, however, I believe can help to improve the quality of the manuscript.

  1. Table 2: Briefly discuss why the elastic modulus of the steel decrease with decreasing the solution pH.
  2. Line 223: Discuss more the origin of the secondary cracks and how they affect the overall ductility of the steel because of SCC.
  3. Figure 5: Show the Nyquist graphs with the same scale at both the x and y-axis (similar Zi and Zr scales).
  4. Lines 249-250: T0, EZ, YS, UTS, and BF should be introduced earlier because they have been shown in Figure 2.
  5. Equations 9 and 10: Check the subscript for H2 and H2CO3. Also, check the potential values.
  6. Line 419: What were the Ecorr values at different pHs?
  7. Discuss the reproducibility of the results.
  8. My main comment on this manuscript is about the lack of novelty. The effect of pH on the SCC has been extensively studied, as the authors have summarized in the Abstract. I highly recommend rewriting your nitch clearly explaining the gap in the literature.

Author Response

Dear reviewer 1 thanks so much for your comments. 

I can tell you that all comments were answered in order to improve the paper

Reviewer 2 Report

This is a well written paper, with good language and reasonable content design. However, there are still some problems or shortcomings that need to be clarified and modified before publication. Some detailed comments are as follows:

  1. Authors need to clarify the pretreatment procedures of the experimental solution. Because the trapped liquid under disbanded coatings, such as NS4 solution, is a strict oxygen free environment, it needs to be deaerated and CO2 treated, which will greatly effect the hydrogen evolution processes. The author needs to clarify whether these procedures exist. If not, its effect on stress corrosion shall be analyzed.
  2. How is EIS continuously tested and how to achieve the time correspondence between EIS results and different elongation states of SSRT?
  3. In Figure 6, all EIS equivalent circuits are not drawn correctly, and the equivalent circuit must be the conduction mode connecting the solution medium and the metal substrate.
  4. In Figure 8, the identified crack initiation position must be supported by the micro morphology obtained under corresponding elongation state.
  5. Some conclusions should be revised after the modifications to above comments.

Author Response

 Dear reviewer 2 thanks so mucho for your comments.

I can tell you that all comments were answered in order to improve the paper. 
Please see the attachment to get the answer of your comments

Reviewer 3 Report

The authors presented scc behavior of x70 which has been studied extensively in the past years. I could not detect the novelty of this research work and the results presented were not groundbreaking to give interest to researchers and readers. However, I would like to share the following with the authors:

The manuscript needs a thorough proofreading to fix some of the sentences.

Figure 2 legend needs enlargement and removal of the horizontal and vertical grid lines.

Figure 3 needs close up magnification to show the features of the micrographs. Labeling such features would illustrate the point of the micrograph. The microcracks and microvoids need to be properly illustrated

The arrows in Figure 7 need to be explained

Author Response

(The authors gave the same response as above.)

Round 2

Reviewer 3 Report

Proofread the manuscript by a professional. 

Author Response

Dear reviewer 1, thanks for your comments and suggestion.

  Suggestion: Proofread the manuscript by a professional.  You can see the manuscript with the corrections made by a professional. The manuscript was attached to this reply.  
